# Identification of Tomato microRNAs in Late Response to *Trichoderma atroviride*

**DOI:** 10.3390/ijms25031617

**Published:** 2024-01-28

**Authors:** Rocío Olmo, Narciso M. Quijada, María Eugenia Morán-Diez, Rosa Hermosa, Enrique Monte

**Affiliations:** Institute for Agribiotechnology Research (CIALE), Department of Microbiology and Genetics, University of Salamanca, 37185 Villamayor, Salamanca, Spain; rol@usal.es (R.O.); nmq@usal.es (N.M.Q.); me.morandiez@usal.es (M.E.M.-D.); rhp@usal.es (R.H.)

**Keywords:** tomato, *Trichoderma*, RNA sequencing, microRNA, miRNA target, priming, quantitative PCR

## Abstract

The tomato (*Solanum lycopersicum*) is an important crop worldwide and is considered a model plant to study stress responses. Small RNAs (sRNAs), 21–24 nucleotides in length, are recognized as a conserved mechanism for regulating gene expression in eukaryotes. Plant endogenous sRNAs, such as microRNA (miRNA), have been involved in disease resistance. High-throughput RNA sequencing was used to analyze the miRNA profile of the aerial part of 30-day-old tomato plants after the application of the fungus *Trichoderma atroviride* to the seeds at the transcriptional memory state. Compared to control plants, ten differentially expressed (DE) miRNAs were identified in those inoculated with *Trichoderma*, five upregulated and five downregulated, of which seven were known (miR166a, miR398-3p, miR408, miR5300, miR6024, miR6027-5p, and miR9471b-3p), and three were putatively novel (novel miR257, novel miR275, and novel miR1767). miRNA expression levels were assessed using real-time quantitative PCR analysis. A plant sRNA target analysis of the DE miRNAs predicted 945 potential target genes, most of them being downregulated (84%). The analysis of KEGG metabolic pathways showed that most of the targets harbored functions associated with plant–pathogen interaction, membrane trafficking, and protein kinases. Expression changes of tomato miRNAs caused by *Trichoderma* are linked to plant defense responses and appear to have long-lasting effects.

## 1. Introduction

*Trichoderma* (Ascomycota, teleomorph *Hypocrea*) is a plant-beneficial fungus of high interest in agriculture, as it is marketed as a biological control agent against plant enemies and as a biostimulant, favoring seed germination, plant growth, and adaptation to abiotic stresses [1]. In plant–*Trichoderma* systems, after the colonization of the roots, the fungus provides the plant with long-lasting resistance against biotic and abiotic stresses by balancing the different phytohormone-dependent pathways, a phenomenon known as priming, which provides the plant with a faster and stronger induction of basal resistance mechanisms upon the perception of a later triggering stimulus [2]. *Trichoderma* is initially recognized by the plant through cell-surface pattern-recognition receptors (PRRs) by means of microbe- or damage-associated molecular patterns (MAMP or DAMP) and apoplastic effectors [3,4]. As a result, the plant’s innate defense (MTI, from MAMP-triggered immunity) is activated, it being sufficient to restrict *Trichoderma* proliferation to the apoplast of the epidermis and cortex. It is then that the early defense signaled by the phytohormone salicylic acid (SA) prevents *Trichoderma* access to the vascular bundles [5]. In this location, *Trichoderma* can maintain an intense and sophisticated dialogue with the plant that activates priming and generates metabolic changes in the plant that prevent the development of disease-causing phytopathogens and insect pests [6,7] and can even act as a barrier to prevent access to filamentous pathogens to the vascular system [8].

Unlike *Trichoderma*, pathogens overcome and suppress MTI by translocating effector molecules into the cytoplasm of the host plant cell, where they interfere with defense responses [9,10]. When the cytoplasmic effector is recognized by a nucleotide binding site (NBS) leucine-rich repeat protein (NLR) receptor, a second specific layer of defense known as effector-triggered immunity (ETI) is activated in the plant [11,12]. NLR gene transcription is required to recognize pathogens and achieve an ETI which is detrimental to plant development and growth. MTI and ETI conserve signaling components that communicate their respective pathways, which would explain why, for example, the production of reactive oxygen species is a critical signaling event connecting the two defense layers [13]. The scenario that comes closer to reality is that MTI and ETI are mutually potentiated to activate stronger defenses and that the NLRs are not uniquely activated by their respective intracellular effectors [14,15]. A third layer of immunity occurs when RNA silencing targets pathogenic RNA or DNA by producing small RNAs (sRNAs) that can recognize complementary sequences of the pathogen with an effect on ETI and MTI responses, resulting in reduced or reinforced virulence and defense [16].

sRNAs, 21–24 nucleotides (nt) in length, are a type of small noncoding RNA that have a variety of biological functions which, together with the RNA-silencing machinery, have emerged as important regulators of the reprogramming of gene expression in plant immunity, pathogen virulence, and communications in plant–microbe interactions [17]. Plant endogenous sRNAs are classified as micro(mi)RNAs or small interfering (si)RNAs, playing key roles in disease resistance [18]. It is well established that NLR genes are the most frequent targets of miRNAs involved in plant defense responses [19], as occurs with at least eight families of miRNAs [20]. Plant MIR genes transcribe miRNAs in the form of single stem–loop structures that are processed by the RNAase III DICER-LIKE 1 (DCL1) endonuclease, giving rise to 3′-methylated double-stranded duplexes (miRNA-5p/miRNA-3p) [21,22], in which the strand remaining functional is loaded into an ARGONAUTE (AGO) protein to form the miRNA-induced silencing complex (RISC) [23]. Plant miRNAs can bind to sequences of target mRNAs by a perfect or near-perfect complementary base pairing which causes the cleavage and degradation of mRNA or represses the translation of target mRNA [24,25]. Thus, miRNAs can function as negative post-transcriptional regulators coordinating plant growth and development, and adaptive responses to abiotic and biotic stresses [26]. Most miRNAs are 21 nt long and are generated by DCL1 [27]. By contrast, recently evolved miRNAs are processed by DCL3 or DCL4 (rather than by DCL1) to produce miRNAs of variable length, usually 23–25 nt long (referred to as long miRNAs or lmiRNAs) [28,29]. Furthermore, 22 nt miRNAs can trigger the generation of 21–24 nt secondary siRNAs from their cleaved target mRNAs produced in a phased pattern (phasiRNAs) that negatively regulate target transcripts with roles ranging from defense activation to chromatin remodeling [30].

The energy cost for the plant to maintain the ETI is high. Once the danger is over, it enters a state of priming which, while alleviating the burden of NLR cost, also favors normal growth. In a postchallenge primed state, the plant stores the defense signals in the ‘transcriptional memory’ [31]. miRNA-targeting NLR genes can trigger the production of phasiRNAs, not only to limit fitness costs but also to regulate NLR gene expression [20]. The coordinated emergence of NLR genes and the NLR-targeting miRNAs supports the plant’s need to balance the diversity and function of NLRs with their suppression [20]. 

In the case of *Trichoderma*, after MTI activation, the fungus may increase the level of a second layer of plant immunity by means of an array of apoplastic effector proteins and metabolites [1]. Although there are so far no known *Trichoderma* effector proteins released into the plant-cell cytoplasm which, after being recognized by a given NLR, may activate other types of defenses. Root and leaf proteomes of *Trichoderma*-treated plants have shown an overrepresentation of NLR-like proteins [32,33], and NLR genes have also been reported to be upregulated in such an interaction [34]. So, the third layer of immunity gives sRNAs a central role in this task, as recently described in the expression of miRNAs in Arabidopsis, with wheat and maize leaves of *Trichoderma*-treated plants linked to the induction of systemic defense [35,36,37]. The priming effect activated by *Trichoderma* in the plant is durable and effective in adapting to pathogen life cycles [38,39], but the *Trichoderma*-triggered phytohormone-signaling network alert disappears over time, becoming imperceptible several weeks after the plant has been in contact with the fungus, to save energy when not needed [40]. Thus, *Trichoderma*-primed plants store the responses to multiple events in their transcriptional memory, modulating the transcription of response genes to future stress [1]. 

In the present work, we have analyzed by RNA sequencing the miRNAs of 30-day-old tomato plants in response to the application of *Trichoderma atroviride* to seeds, to know their changes at a transcriptional memory state and the predicted miRNA target genes and to explore their potential functional role in plant beneficial effects associated to *Trichoderma*. 

## 2. Results

### 2.1. Identification and Categorization of miRNAs in Tomato Plants Treated with Trichoderma

Three biological replicates (from now on, “sample”) per condition (untreated -Mock, “C”; and *T. atroviride* T11-treated, “T11”), each one containing three tomato plants, were subjected to sRNA sequencing. After quality control and filtering, an average of 26.4 million reads per sample were obtained (Table 1). A mean value of 6.5 million reads per sample aligned uniquely against the *Solanum lycopersicum* genome, where 8.0% of them aligned against the miRNA candidates (Table 1). 

Overall, 1889 miRNA candidates were identified from a total of 3.3 million reads, aligning to their mature sequence and prediction of miRNA hairpin structures in the genome of tomato (Appendix A). A count matrix output containing the raw counts of all miRNAs identified, as well as additional information that includes the pre-miRNA sequence and its secondary structure, the sequence of the mature miRNA and the sequence of the star miRNA (miRNA*), and the number of reads mapped to miRNA-5p/miRNA-3p (precursor, mature, star, and antisense) sequences for each of the replicates and conditions, is also presented in Appendix A. Reads of 20–24 nt long miRNAs accounted for over 99.99% of the total number of reads that aligned against the miRNA candidates, among which 21 nt miRNA long reads were the ones that showed the greatest transcription overall, as they accounted for most of the sequencing reads (86.6% of all miRNA reads). Because bona fide miRNAs of 24 nt are rare to find in databases, and even though the annotation of miRNAs followed the miR–PREFeR pipeline criteria, a manual detailed review of the data generated by the algorithm was performed in order to improve accuracy and reduce noise when annotating. In our particular case, putative miRNAs with less than 10 read counts aligning to their miRNA* sequence were discarded. We found that each miRNA rises from a single pre-miRNA, which is scattered around the genome (Appendix A). Taking these into account, only 87 out of the first-identified 1889 pre-miRNAs overcame this threshold (Appendix A). The size distribution of these 87 miRNAs, and the amount of reads aligning to them, is shown in Figure 1A. From these miRNAs, nine of them were found exclusively for plants treated with strain T11, six solely for those untreated (C), and 72 for those under both conditions (Figure 1B). A principal component analysis (PCA) based on counts per million (CPM) values of the 87 identified miRNAs shows a separation of C and T11 samples, except for one of the T11 replicates (T11-R1) that clustered close to the control samples (Appendix A).

Of the 87 candidate miRNAs with at least 10 reads aligning to their miRNA* sequence, only those that were transcribed in at least two of the three biological replicates of each condition were considered for the downstream analysis. This threshold reduced the list to 72 miRNAs (Appendix A) which, using the miR–PREFeR algorithm, allowed us to verify that 40 of them harbored a sequence that matched miRNAs previously annotated at the miRBase repository. The 32 remaining sequences did not return any match and, thus, were labeled as “novel miRNAs” (Appendix A). Sequences and characteristics of these 32 novel miRNAs were predicted by using sRNAtoolbox and are shown in Appendix A, reflecting that 78% of these novel miRNAs were 24 nt long and, as stated above, are rare to find in databases. Overall, the finding of novel miRNAs in *S. lycopersicum* has provided enriched insight into the plant miRNA repertoire. The functions of these novel miRNAs need to be further demonstrated.

### 2.2. Differential Expression of miRNAs in T11-Treated Tomato Plants

After normalization of the reads as CPM and an independent DE analysis of the miRNAs (Appendix A), we identified that only 10 miRNAs were DE (Wald test *p*-value < 0.05 and *p*-value adjusted using Benjamini and Hochberg’s method (*p*adj) < 0.1) in plants from C and T11 conditions (Table 2). Five of those DE miRNAs were downregulated, while the other five were upregulated in tomato plants treated with *T. atroviride* T11 (Table 2 and Appendix A).

The presence of paralogous *MIR* loci that produce identical or nearly identical mature miRNAs is frequent within the genome [41,42], and, so, they are grouped together into families. In this case, miRNA_607 and miRNA_608 have identical mature sequences that correspond with the conserved miRNA5300 (miR5300) and, hence, were grouped in the same MIR5300 family (Table 2). The results showed the DE of several microRNAs involved in the regulation of stress response, such as miR166a, miR398-3p, miR408, miR5300, miR6024, and miR6027-5p.

Interestingly, 3 of these 10 DE miRNAs were novel. One of them was exclusively found in the control plants (novel miR1767), another one was only found in T11-treated plants (novel miR275), and the third was identified in both conditions (novel miR257; Appendix A). The sequence and characteristics of these three novel miRNAs predicted with sRNAtoolbox are shown in Table 3, their precursor sequences are in Appendix A, and the stem–loop hairpin secondary structures, are represented in Figure 2.

The DE analysis aimed to highlight those miRNAs that showed a differential level of expression (Wald test *p*-value < 0.05 and *p*-value adjusted using Benjamini and Hochberg’s method (padj) < 0.1) between the control and the T11-treated plants, regardless of their level of expression, towards their inclusion into further validation analyses. Fold-change values for the DE miRNAs were validated by a TaqMan^®^ assay based on real-time quantitative PCR (qPCR) (Figure 3). Six miRNAs were selected for validation, with four downregulated and two upregulated in the T11-treated plants with regard to the C condition. These were, in turn, representative of the expression differences in both conditions (in terms of CPM), such as high transcription (miR166a), low transcription (novel miR1767), and intermediate transcription (miR6027-5p). A qPCR was performed after cDNA synthesis with the same RNA samples used for the construction of the six libraries and resulted in a high validation rate of the sequencing results. With the sole exception of the novel miR1767, which might be due to the low expression encountered for this miRNA, similar expression trends were observed for the miRNAs evaluated by RNA sequencing and qPCR (Figure 3). miRNA166a and miR398-3p showed significant differences in transcription for plants under C and T11 conditions in the qPCR analysis. Therefore, 83% of selected miRNAs were validated by qPCR.

### 2.3. Prediction of miRNA Target Genes and Their Biological Function

The prediction of the genes that are targets of the identified miRNA was performed by using psRNATarget against their reference *S. lycopersicum* genome (genome version SL4.0 and annotation version ITAG4.0). Plants treated with T11 showed 596 putative target genes for the upregulated miRNAs and 532 putative target genes for the downregulated ones. Over 84% of the target genes identified (*n* = 945) were predicted to be negatively regulated by miRNAs in a miRNA cleavage manner, while the remaining 16% (*n* = 183) might be translationally repressed by the miRNA’s mediation (Appendix A).

The target’s functional prediction based on the KEGG metabolic pathways database showed that most targets harbored functions related to main broad cell functions, including gene information and processing (*n* = 411, 23.4% of all targets), signal transduction (*n* = 202, 11.5%), and metabolism (*n* = 159, 9.1%) (Figure 4A,B). Strikingly, when deepening the pathway’s hierarchy, we found that those proteins associated with plant–pathogen interaction functions were the most abundant targets overall, highlighting the potential role of the DE miRNAs identified here to be involved in plant–microorganism crosstalk.

Moreover, Figure 5 reveals that those miRNAs that were significantly upregulated in the T11 condition (miRNA5300, miRNA6024, miRNA9471b, novel miRNA257, and novel miRNA275) were the ones that significantly affected target genes associated with plant–pathogen interaction functions, highlighting the potential role of *Trichoderma* to induce the transcription of miRNAs involved in the molecular dialogue with the plant.

As an example, in Table 4, we expose a selection of genes that are involved in the response to abiotic and biotic stress and genetic information processing.

To examine the correlation between the targets and their corresponding miRNAs, primers were designed to evaluate the transcription levels of five selected targets by qPCR (highlighted in bold in Table 4). These targets were selected because they have been described as playing important roles in plant–pathogen interactions. As shown in Figure 6, some negative correlations were found between the expression levels of the target genes and their corresponding miRNAs in the control and T11-treated plants, implying that miRNA-mediated mRNA silencing occurs when tomato is treated with *T. atroviride* T11. *MIR166a* was downregulated in T11-treated tomato plants. Accordingly, its target gene, encoding a Class III homeodomain-leucine zipper protein (HD-ZIP III) (protein ID Solyc11g069470.3.1), was upregulated, while *MIR5300* was upregulated in T11-treated tomato plants and its predicted target gene encoding an O-fucosyltransferase family protein (AT1G52630-like protein; Solyc03g116550.4.1) was downregulated (Figure 6).

## 3. Discussion

For many years, studies for fruit development have had the tomato as a research model, mainly since the completion of its genome [43], which has made it an excellent system to study molecular plant–microorganism interactions, including those with *Trichoderma* [40,44]. There is sufficient information on the importance of sRNAs in the regulation of plant responses to biotic and abiotic stresses [20,45]. However, under the sRNAs, a plethora of diverse RNA molecules 20–30 nt long that have emerged as major regulators in plants still have little-known roles in plant–*Trichoderma* interaction. In recent years, miRNAs from *Trichoderma*-treated plants have begun to be cataloged [35,37,46], although more research is still needed to understand this interaction and the role of the third layer of plant immunity in *Trichoderma*-induced priming. In this study, we aimed to explore the miRNAs in tomato plants for a long time (30 days) after inoculation with *T. atroviride*, when the phytohormonal signaling of the plant defenses is quiescent [47] and the plant is in a state of transcriptional memory [1]. Thus, we have performed RNA sequencing with three biological replicas of tomato-leaf samples from untreated plants and plants from seeds coated with *T. atroviride* T11. Previous works have analyzed the expression profiles of miRNAs in plants at very short times after *Trichoderma* application [35,37], while we have wanted to explore which miRNAs related to plant defense are differentially expressed when priming is dormant and only a few genes are differentially transcribed in *Trichoderma*-treated plants compared to the control [40] and at a transcriptional memory state. Thus, we have performed RNA sequencing with three biological replicas of 30-day-old tomato-leaf samples from untreated plants and plants from seeds coated with *T. atroviride* T11.

Many algorithms are available for plant miRNA prediction; yet, the existing annotations harbor many discrepancies, with 24 nt sequences more prone to be falsely annotated as miRNAs [22]. Based on the miR–PREFeR’s algorithm, we identified a total of 1889 potential miRNAs with a hairpin structure obtained from the tomato genome, of which 75.6% were 24 nt long and only 9.1% were annotated as 21 nt miRNAs. The latter is the most described size for miRNAs [22] and accounted for most of the reads in this study. We found a high percentage of 24 nt miRNAs that are not identified as the canonical size, although there are also many described in both the model and cultivated plants, such as *Lotus japonicus* [48], rice [49,50], and apple [51]. Because rigorous prediction and annotation of miRNAs is one of the challenges in this type of study, we removed all miRNA candidates that harbored less than 10 quality-filtered sequencing reads aligning to the miRNA* sequence, thus narrowing the list to 87 miRNAs. Additionally, only miRNAs present in at least two of three biological replicates of a condition were considered, reducing the total number to 72 miRNAs. Thirty-two out of the final 72 potential miRNAs selected were found to be novel. The 44.4% of novel tomato miRNAs found in response to *T. atroviride*, agrees with the 25–40% obtained for novel miRNAs detected in tomato plants under biotic stress by *Phytophthora*, *Botrytis,* or *Pseudomonas*, but the proportion is lower than that observed in tomato under abiotic stresses, such as heat, drought, or chemical treatment [52,53]. Among the 72 preselected miRNAs, only 10 were DE (*p*adj < 0.01, according to [53]) in T11-treated plants. Other works detected higher numbers of DE miRNAs in tomato plants, but at much shorter time points (between 3 and 96 h) in response to *Phytophthora* [52] or under biotic and abiotic stresses at time points of less than one week [53]. Particularly, Lopez-Galiano et al. (2019) [53] detected 17 and 3 DE miRNAs in response to *Pseudomonas* and *Botrytis* inoculation, respectively. Therefore, taking into account that we have considered only the tomato response to T11 at one longer sampling time, it seems that 10 DE microRNAs should not be considered a low figure.

Given the disparity of one of the three samples from the T11 condition detected in the PCA (Appendix A), which clustered closer to the control samples, we have tried to overcome this difficulty by confirming RNA-sequencing DE results using a qPCR analysis of the three biological replicates of each condition for the representative 6 out of 10 DE miRNAs. It is not rare to find 100% matched patterns in comparisons between RNA sequencing and qPCR for such studies with miRNAs [52]. However, in our case, the validation rate was 83%. Differences in sensitivities between the two techniques have also been observed for the tomato-plant responses to different biotic and abiotic stresses [53]. Among the seven known DE miRNAs, there were six (miR166a, miR398, miR408, and miR6027 were downregulated and miR5300 and miR6024 were upregulated) that had previously been linked to stress responses and development in tomato plants [54,55,56].

The ancient miR166 family is highly conserved among plants and associated with HD-ZIP III transcription factor family genes [57]. Tomato tolerance to *Phytophthora infestans*, tomato leaf curl New Delhi virus (ToLCNDV), and cold stress have been shown to be linked to the upregulation of miR166 [58,59,60]. Conversely, we have seen that T11-treated plants showed reduced levels of miR166, while its HD-ZIP III target gene was induced. Our results are consistent with previous works showing that miR166 suppression raises the levels of its corresponding HD-ZIP III target, with increased tolerance to yellow leaf curling virus in tomato [61], responses of potato plants to *P. infestans* [62], and Arabidopsis to the nematode *Meloidogyne incognita* [63]. However, HD-ZIP III transcription factors are not only involved in defense but also in plant growth by regulating the transport and flow of auxins that condition developmental responses [64].

miR398 is a highly conserved miRNA that is widespread in angiosperms [29] and is considered a master regulator of developmental and environmental stress responses [53,65]. Although miR398 has been described as induced by *P. infestans* inoculation in the tomato [66], we have seen that it is downregulated in T11-treated plants, as occurred in the typical responses of tomato plants tolerant to abiotic stresses [45,67]. Moreover, the expression of a putative miR398 target gene, encoding an NLR protein, did not show expression changes in T11-treated plants, suggesting that this type of immune response does not seem to be activated 30 days after the application of *T. atroviride*. miR408 is an ancient and highly conserved miRNA that is involved in the regulation of plant growth, development, and stress response [68]. In this line, miR408 is also downregulated in T11-treated plants while its upregulation has been associated with responses to ToLCNDV [69] and its decline has been related to drought tolerance of the aerial parts of tomato plants [67]. miR6027 is one of the master regulators of NLR proteins in Solanaceae plants [70,71], and its downregulation in T11-inoculated plants would be associated with increased defense. In this sense, a previous study associated miR6027 downregulation caused by *P. infestans* in tomato plants with the expression of *NLR* genes involved in ETI and, thus, resistance to this pathogen [66]. However, the predicted PRR target of miR6027, a leucine-rich repeat receptor protein kinase gene that was tested for expression, did not show changes between the control and T11-treated plants. This would indicate that plant defenses would be dormant after one month of *T. atroviride* application. According to previous studies in tomato plants, downregulated miRNAs, such as miR5300 and miR6024, are related to tolerance to *Botrytis*, *Fusarium, Alternaria,* and *Phytophthora* pathogens [45,72,73,74]. Moreover, the miR6024–NLR interaction facilitates necrotrophic pathogenesis by *Alternaria solani* in the tomato [75]. In our study, the upregulation of miR6024 was not accomplished to transcriptional changes in the NLR target, possibly because, if it is not necessary, the plant saves defense costs. The existence of upregulated miR5300 in T11-treated plants and the negative coregulation of its predicted target, an *O-fucosyltransferase* gene involved in the glycosylation of PRRs and the plant growth repressors, DELLA proteins [76], is consistent with not activating defenses when they are not needed and not inhibiting plant growth, a behavior compatible with the beneficial effects of *Trichoderma* on plants [6].

Considering the functional annotation of the genes that are targets of DE miRNAs (according to the KEGG database), we identified that those plants inoculated with *T. atroviride* significantly affected the transcription of miRNAs affecting genes associated with plant defense, trafficking across the membrane where PRR receptors are located, signal-transduction pathways, and phytohormone signaling networks. This functional characterization reinforces the role of these miRNAs in the priming of defenses activated by *Trichoderma*, regardless of whether the plants’ responses are dormant or stored in the transcriptional memory after several weeks of being triggered.

## 4. Materials and Methods

### 4.1. Tomato Plants, T11 Inoculation, and Sample Collection

A *T. atroviride* T11-conidial suspension was harvested from sporulated PDA plates with 10 mL of sterile water and filtered through glass wool to remove mycelia. The concentration of conidia was calculated by using a hemocytometer chamber. A total of 40 tomato seeds (*Solanum lycopersicum* ‘Marmande Raf’; EuroGarden, Valencia, Spain) per condition were surface-disinfected and coated with 1 mL of a 10^8^ conidia/mL suspension of T11 strain. Control seeds were dipped in 1 mL of sterile water. Coated seeds were air-dried in open Petri dishes under aseptic conditions in a laminar air-flow hood. Seeds were sown in plastic pots (9 × 9 × 10 cm) containing a 3:1 mixture of substrate (50% clay–50% cocopeat; 0.8 kg/m^3^ of NPK 14-16-18 [N-P_2_O_5_-K_2_O], and pH 6–6.5) and vermiculite previously two times sterilized in autoclave.

Three initial seeds were sown per plot until their germination after which only one was left per pot. Plants grew for 30 days in a greenhouse under controlled conditions of humidity (75%), a 16 h photoperiod, a temperature of 18–28 °C, and a conventional irrigation regimen with tap water. The experimental setup was carried out with nine tomato plants per condition in a randomized design. Conditions were assigned as follows: control (untreated plants) and T11 (plants treated with *T. atroviride* T11 strain).

Two opposite leaflets of the stems on the second set of true leaves were detached per plant from three plants per biological replicate, placed on a Falcon tube, and frozen in liquid nitrogen for further RNA analysis. Three biological replicates were considered per condition for sRNA analysis.

### 4.2. RNA Extraction and Sequencing

Total RNA was extracted from the leaves of tomato plants with TRIzol Reagent (Invitrogen Life Technologies, Carlsbad, CA, USA) following the manufacturer’s instructions, including a final treatment step with DNAse (Invitrogen Life Technologies), and the samples were purified with the GeneJet RNA Cleanup and Concentration Micro Kit (Qiagen, Hilden, Germany). Quality control of the total RNA profile was evaluated with the Qubit Fluorometer (Invitrogen Life Technologies) and the RNA 6000 Nano Kit as the 2100 Bioanalyzer Instrument (Agilent, Santa Clara, CA, USA). All samples were also run in a 1% agarose gel for visual quality evaluation. A total of 4 µg of RNA per sample were sent at a concentration of 200 ng/µL to ADM Lifesequencing (Valencia, Spain) for library construction and sequencing. Sequencing libraries were prepared by Illumina protocol based on the TruSeq Small RNA library kit, and, subsequently, Illumina sequencing-by-synthesis technology was used to generate the libraries, yielding over 185 million 75 bp single-end raw reads overall and a median of ca. 27.8 million reads per sample (Table 1).

### 4.3. Data Quality Control and Identification of Known and Novel miRNA Candidates

Quality control of the sequencing data and the identification of known and novel miRNA candidates were performed by Sequentia Biotech S.L. (Barcelona, Spain) as follows. The raw sequencing data quality was assessed by using FastQC [77]. Sequences with a Phred quality score below 30 and a length below 15 bp were removed, and sequencing artifacts and adapters were removed by using BBDuk [78]. Quality filtering resulted in ca. 173.8 million reads overcoming the filtering overall (median of 26.4 million reads per sample, Table 1). Reads that matched to noncoding RNAs, such as ribosomal RNA (rRNA), transfer RNA (tRNA), small nuclear RNA (snRNA), or small nucleolar RNA (snoRNA), were also discarded.

The reads that overcame the quality controls described before were subjected to the miR–PREFeR pipeline [79] and the miRbase database [80] in order to identify miRNAs by aligning the reads to the *S. lycopersicum* assembled genome version ITAG4.0. The miR–PREFeR pipeline includes SAMTools [81] to generate candidate regions according to the abundance of each unique read. The hierarchical mode was enabled; thus, all reads of a given library can align only once to one annotation group. Following this initial step, the identified regions were folded using the RNAfold program with sRNAtoolbox from the ViennaRNA package 2.0 [82] to detect stem–loop structures that will be examined using plant miRNA annotation criteria [42]. The pipeline was run under default settings and using the “-f” parameter to filter out unmapped alignments in the output. In order to discard potential false positives, miRNA candidates were considered only when there were reads aligning to them at least in two out of the three samples per condition. Moreover, miRNA candidates that showed less than 10 read counts against their miRNA* sequence were also discarded.

### 4.4. Prediction of miRNA Target Genes and KEGG Function Analysis

Target mRNA candidates for each miRNA were predicted using psRNATarget [83] under a default scoring schema V2 (penalty for the G:U pair set at 0.5; mismatches allowed in the seed region set at 2–13; and maximum energy to unpair the target site 25). The functional profile of the predicted RNA targets was assessed by alignment of their proteins against the Kyoto Encyclopedia of Genes and Genomes (KEGG) [84] and using Diamond v2.0.7 [85], with a minimum amino acid sequence identity and alignment query coverage of 70% and a maximum e-value of 1 × e^−5^.

### 4.5. Statistical Analysis of Sequencing Data and Data Visualization

Counts per million (CPM) normalization values were calculated for each miRNA and sample according to the next formula:$$CPM=\frac{number\;of\;miRNA\;reads}{total\;number\;of\;RNA\;reads}\times 10^{6}\;$$

DE gene analysis was performed in R environment v4.0.3 [86] by using DESeq2 v1.26.0 [87] and the unnormalized counts of the miRNA data. miRNAs with less than 10 mapped reads overall were discarded for DE gene analysis. Raw values of *p* were adjusted for multiple testing using Benjamini and Hochberg’s method [88], which assesses the false discovery rate. Gene transcripts with an adjusted *p* < 0.1 were considered to be differentially transcribed between the two experimental conditions investigated, according to López-Galiano et al. [53].

A principal component analysis (PCA) based on CPM calculations was performed by using Factoextra [89]. Bar plots, donut charts, and PCA were visualized by using ggplot2 [90], reshape [91], and ggpubr [92]. A Venn diagram was computed and visualized by using ggVennDiagram [93]. Heatmaps were calculated and visualized by using pheatmap [94].

The sequence data were deposited in the NCBI Short Read Archive (SRA) with the Submission ID: SUB13960687, BioProject: PRJNA1037012, and Biosamples: SAMN38155784 to SAMN38155789.

### 4.6. Validation of miRNAs and Their Target Genes by qPCR

The expression levels of selected DE miRNAs in the libraries were analyzed by qPCR using a TaqMan small RNA assay. The probes used were designed and synthesized by a Custom TaqMan small RNA assay (Thermo Fisher Scientific, Waltham, MA, USA) for novel miR1767, miR166a, miR398-3p, miR5300, miR6024, and miR6027-5p. The three independent RNA samples per condition used for RNAseq libraries were used as templates for cDNA synthesis. Total RNA (1–10 ng) was reverse transcribed using a TaqMan microRNA Reverse Transcription Kit (Thermo Fisher Scientific) in a final reaction volume of 15 µL. qPCR was performed using TaqMan Universal Master Mix II (no UNG; Thermo Fisher Scientific) on a StepOnePlus Real-Time PCR System (Applied Biosystems, Foster City, CA, USA), following the manufacturer’s protocol. Each independent cDNA template was run in triplicate. Ct data were normalized to the expression of reference gene *U6 small nuclear RNA* (*U6 snRNA*), and the relative expression was calculated using the 2^−ΔΔCt^ method [95].

Validation of the transcript level for predicted miRNA target genes was performed by qPCR. The three independent RNA samples per condition used for RNAseq libraries were also used for cDNA synthesis. cDNA synthesis, PCR mixtures, and amplification conditions were as previously described [44]. qPCR was performed with a StepOnePlus Real-Time PCR System (Applied Biosystems), using KAPA SYBR FAST (Biosystems, Buenos Aires, Argentina), and the described primer couples are listed in Appendix A. For each sample, three biological replicates (with three technical replicates each) were analyzed. Ct data were normalized to the expression of the reference housekeeping gene *Actin* (*ACT*) and relative expression was calculated using the 2^−ΔΔCt^ method [95].

qPCR data were analyzed by Student’s *t*-test for statistically significant differences (*p* < 0.05).

## 5. Conclusions

sRNAs constitute the third layer of plant defense affecting ETI and MTI responses, but the role of miRNAs compared between interactions with pathogenic and beneficial microorganisms is still being deciphered. Moreover, the responses to beneficial microorganisms after several weeks of interaction with the plant, such as the one carried out in the present study, is a subject that needs further exploration. RNA sequencing of 30-day-old tomato plants derived from seeds that were inoculated with *T. atroviride* T11 served to identify 10 DE miRNAs, of which three were novel and seven had previously been associated with defense responses and development in tomato plants. Tomato plants inoculated with *T. atroviride* T11 showed that the downregulation of miRNAs, such as miR398, miR408, and miR6027, and the upregulation of miR6024 did not lead to modifications in the expression of their PRR or NLR targets, which would indicate that, pending more detailed functional studies on miRNA–target pairs, at a transcriptional memory-state level the defenses would be switched off. In addition, miRNAs show canonical behavior with their respective targets, with a decrease in the miRNA166 level and its upregulated target HD-ZIP class III and an increase in the miRNA5300 level and its downregulated target O-fucosyltransferase, which could both indicate that growth promotion in plants treated with T11 would remain active one month after inoculation of this beneficial fungus.

## Figures and Tables

**Figure 1 ijms-25-01617-f001:**
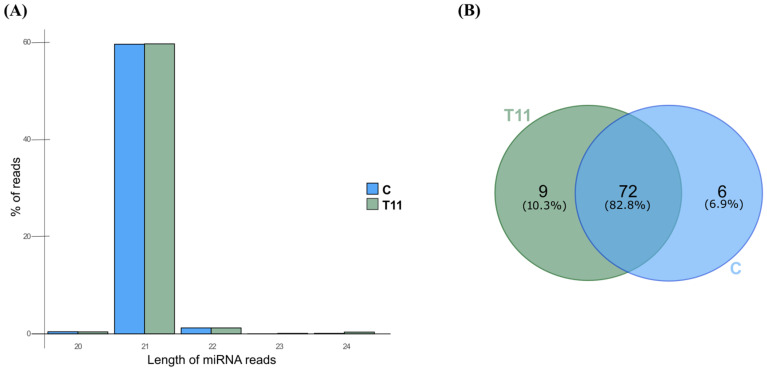
General analysis in *Trichoderma atroviride* T11-treated (T11) *Solanum lycopersicum* or untreated (C) plants libraries. (**A**) Length distribution and abundance of the miRNAs that showed at least 10 read counts against their miRNA* sequence in *S. lycopersicum* plants treated (T11) or not (C). The percentage is calculated over the total number of reads that aligned to the miRNA candidates identified in plants from both conditions (including those with less than 10 read counts against their miRNA* sequence and/or those that were solely transcribed in one sample per condition). The length of the miRNAs identified ranged from 20–24 nt; (**B**) Venn diagram showing the number of miRNAs (that displayed at least 10 read counts against their miRNA* sequence) identified in both conditions, solely in T11 (green) or in untreated control plants (blue).

**Figure 2 ijms-25-01617-f002:**
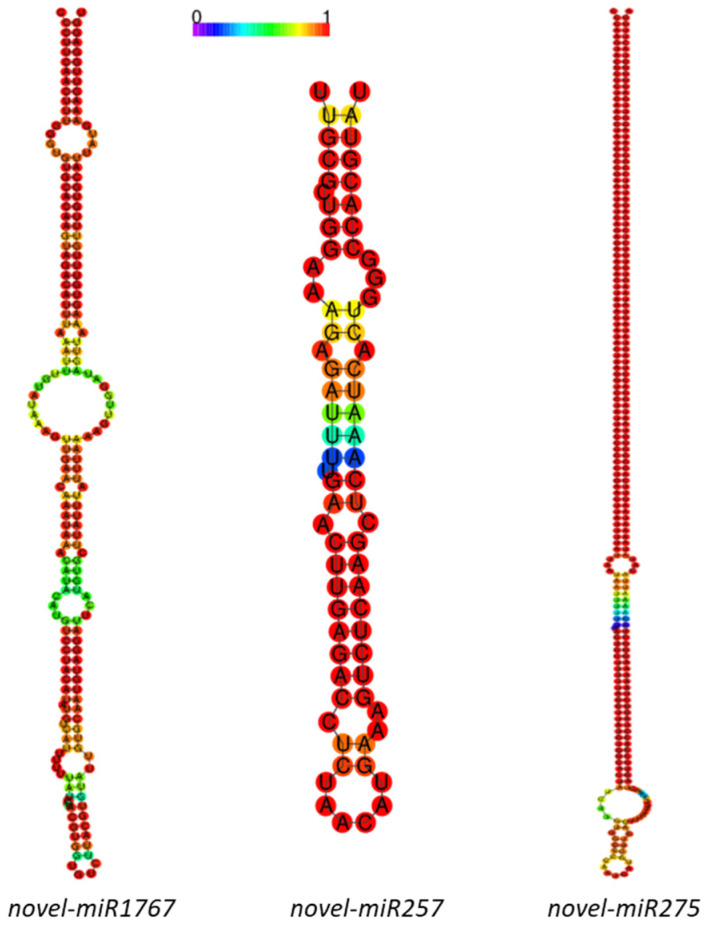
Precursor hairpin structures of three differentially expressed novel *Solanum lycopersicum* miRNAs. Mature miRNA sequences are shown. The color gradient indicates base-pair probabilities.

**Figure 3 ijms-25-01617-f003:**
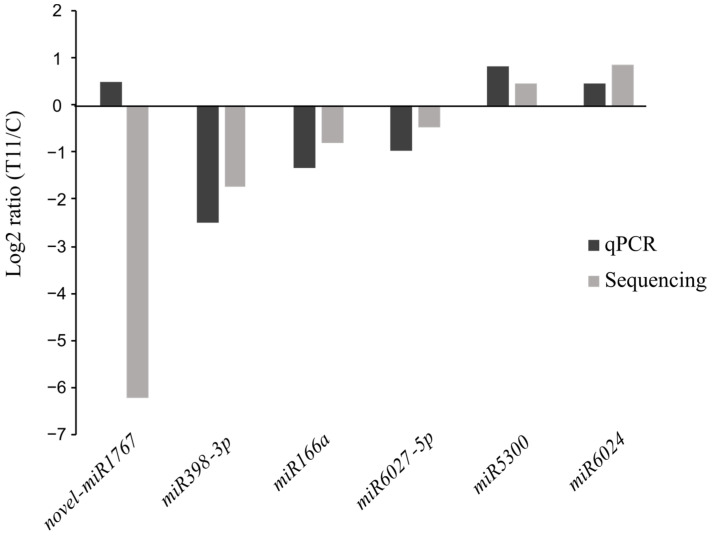
Real-time quantitative PCR (qPCR) analysis of differentially expressed miRNAs. miRNA expression in *Trichoderma atroviride* T11-treated to untreated control plants by RNAseq (gray) and qPCR (black). Negative values—downregulated in T11-treated plants; positive values—upregulated in T11-treated plants.

**Figure 4 ijms-25-01617-f004:**
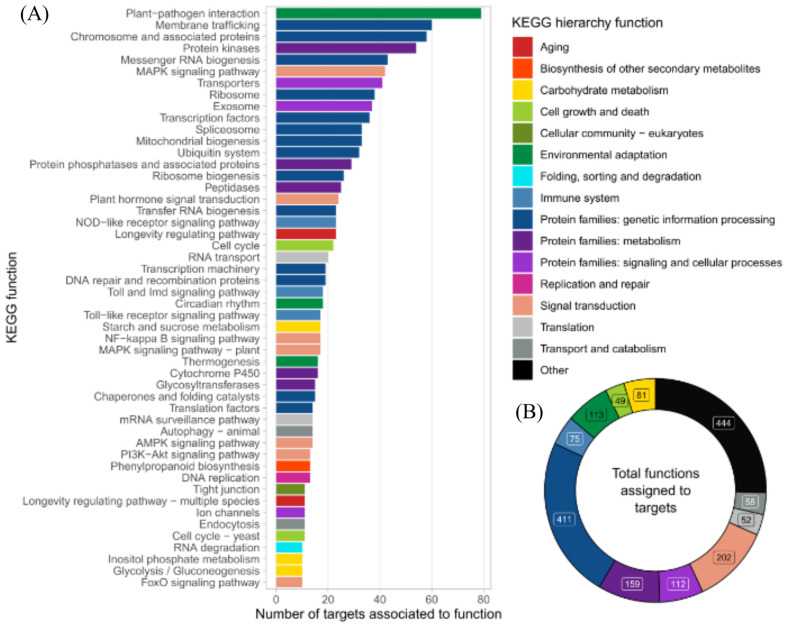
KEGG analysis of target transcripts regulated by differentially expressed (DE) miRNAs. (**A**) Number of targets identified for the DE miRNAs associated with the different KEGG functions, sorted for function associated with a greater number of targets to lower. (**B**) Number of targets identified for the DE miRNAs overall is associated with broader KEGG hierarchy functionalities.

**Figure 5 ijms-25-01617-f005:**
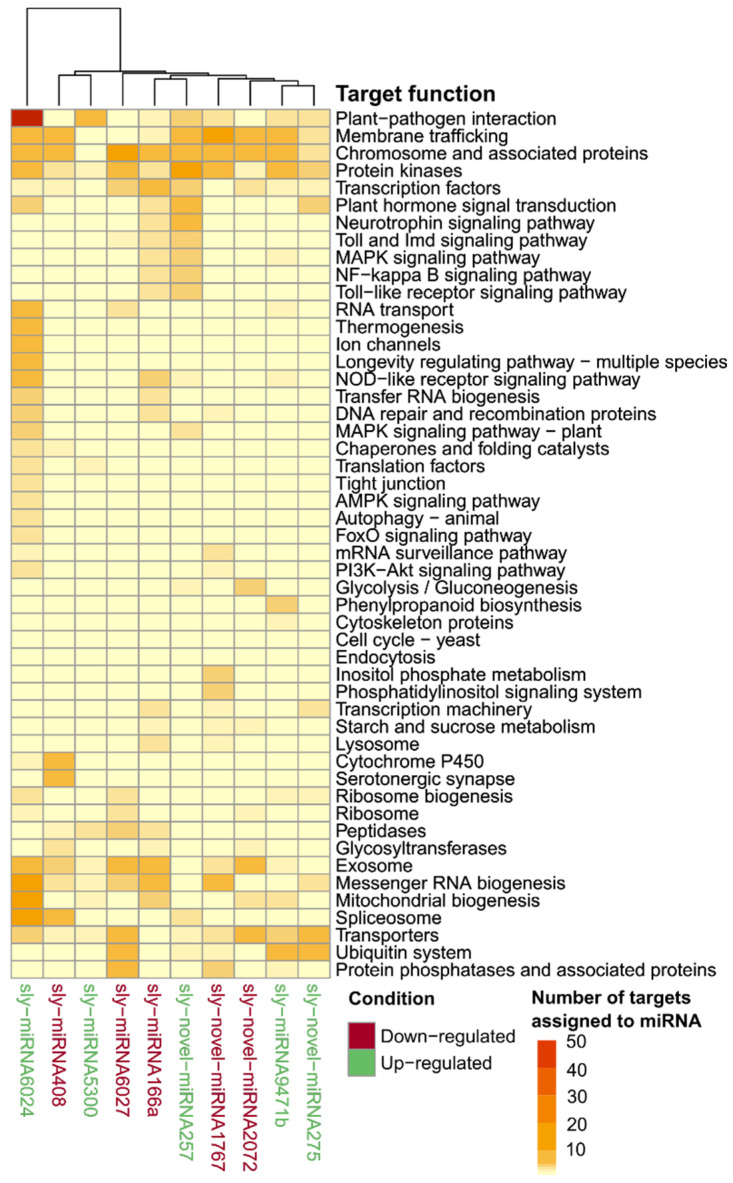
Heatmap showing the number of targets (from light yellow, low, to dark red, high) that were associated with the different differentially expressed (DE) miRNAs (labeled in green or red if they were significantly DE in T11 or control conditions, respectively).

**Figure 6 ijms-25-01617-f006:**
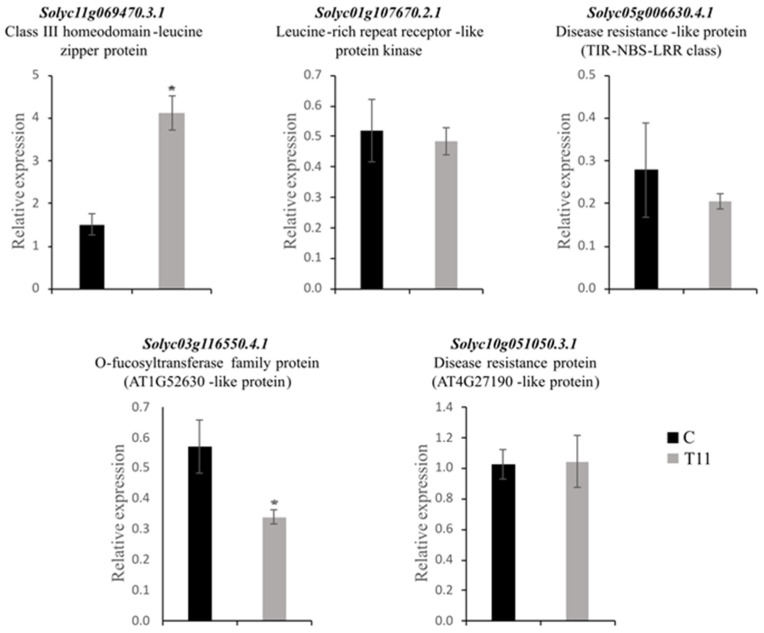
Real-time quantitative PCR (qPCR) analysis of five targets of differentially expressed (DE) miRNAs in *Trichoderma atroviride* T11-treated (T11) *Solanum lycopersicum* plants or untreated (C) plants libraries. The housekeeping gene *Actin* was used as the internal control, and error bars represent the SEM of three replicates. Asterisks indicate a significant difference as determined by Student’s *t* test (*p* < 0.05).

**Table 1 ijms-25-01617-t001:** Summary of total reads for miRNAs analyzed in untreated (C) and *Trichoderma atroviride* T11-treated (T11) plants. Three biological replicates per condition were used for miRNA analysis (noted as R1, R2, and R3). The minimum length of the reads after trimming was set to 15 bp with a Q score > 30. Reads that aligned in multiple positions of the *S. lycopersicum* genome were discarded for downstream analysis.

No. Library	Biological Replicate	Number of Raw Reads	Number of Reads after QC	Overall Alignment Rate (%) against *S. lycopersicum* Genome (from QC Reads)	Number of Reads That Aligned Once against *S. lycopersicum* Genome	Number of Reads Aligned against miRNA Candidates
1	C_R1	31,588,579	30,037,348	98.5	6,491,685	692,721
2	C_R2	18,479,108	17,586,087	98.4	4,280,339	315,717
3	C_R3	32,775,529	31,349,341	95.6	6,584,081	539,053
4	T11_R1	24,024,180	22,844,527	98.6	4,976,203	320,623
5	T11_R2	17,523,164	16,525,822	98.1	5,066,436	278,130
6	T11_R3	61,256,777	55,450,480	98.1	11,635,432	1,165,112

**Table 2 ijms-25-01617-t002:** List of differentially expressed (DE) novel (gray) and conserved (black) miRNAs in samples from untreated (C) and *Trichoderma atroviride* T11-treated (T11) tomato plants under the miR–PREFeR algorithm. Red downregulated and green upregulated in T11-treated plants. MIR family names group those miRNAs that showed a similar mature miRNA sequence. The number of the miRNA_ID code was used to assign the family’s name in the case of novel miRNA. Statistical analysis was performed with R using the package DEeq by applying the log2FC (only those with a padj < 0.1 are shown). CPM stands for “counts per million”, while lfcSE stands for “log2FC standard error”.

miRNA_ID	miRNA_Name	MIR Family Name	Mean CPM Control	Mean CPM T11	log2FC	lfcSE	*p*adj (<0.1)
miRNA_1767	novel miR1767	MIR1767	0.46	0.00	−6.243	1.673	0.00296
miRNA_237	miR408	MIR408	2.36	0.67	−1.896	0.478	0.00141
miRNA_2072	miR398-3p	MIR398	2.55	0.78	−1.745	0.679	0.06613
miRNA_965	miR166a	MIR166	5901.95	3597.31	−0.805	0.145	1.99 × 10^−6^
miRNA_1734	miR6027-5p	MIR6027	185.79	141.93	−0.471	0.091	6.66 × 10^−6^
miRNA_1908	miR9471b-3p	MIR9741	57.08	83.67	0.428	0.142	0.02008
miRNA_607	miR5300	MIR5300	17.26	25.20	0.454	0.132	0.00553
miRNA_608	miR5300		17.26	25.20	0.454	0.132	0.00553
miRNA_181	miR6024	MIR6024	4.08	7.96	0.856	0.245	0.00553
miRNA_257	novel miR257	MIR257	0.49	1.25	1.169	0.432	0.04789
miRNA_275	novel miR275	MIR275	0.02	0.00	10.315	3.307	0.01574

**Table 3 ijms-25-01617-t003:** Main features of the three DE novel *Solanum lycopersicum* miRNAs.

miRNA ID	miRNA Name	miRNA Mature Sequence	Length	% GC	*S. lycopersicum* Chromosome	miRNA Chromosome Start Position	miRNA Chromosome End Position	Strand	Minimum Free Energy (kcal/mol)
miRNA_1767	novel miR1767	CUUCAACUUUGGGUGUGCACAAGU	24	45.83%	11	2,825,245	2,825,268	-	−61.8
miRNA_257	novel miR257	AAAGAGAUUUUGAACUUGAGACCU	24	33.33%	1	88,918,167	88,918,190	-	−24.1
miRNA_275	novel miR275	CUCUGAGAUUUCGGGCAUAGGUU	23	47.83%	2	19,634,044	19,634,334	-	−222.5

**Table 4 ijms-25-01617-t004:** Predicted target genes of six selected differentially expressed (DE) miRNAs from the study. “Exp” refers to the expected complementarity between a miRNA and its targets (the lower the better, ranging from 0 to 3.5), while “UPE” stands for the maximum energy to unpair the target site (the lower the better). Further information includes miRNA length, target start and end of complementarity with the miRNA, miRNA, and target aligned fragments, kind of inhibition of expression (either cleavage guidance or translation arrest), KEGG orthologue ID (“NA”, no KEGG orthologue identified under the alignment cutoffs of the protein against the KEGG database), and target description according to psRNATarget. Red boxes show downregulated, and green show upregulated in T11-treated plants. Target genes in bold were validated by qPCR.

miRNA	Target Accession	Exp	UPE	miRNA Length	Target Start	Target End	miRNA Aligned Fragment	Target Aligned Fragment	Inhibition	KEGG Orthologue ID	Target Description
** *miR166a* **	**Solyc11g069470.3.1**	**1**	**23.445**	**21**	**653**	**673**	**UCGGACCAGGCUUCAUUCCCC**	**CUGGGAUGAAGCCUGGUCCGG**	**Cleavage**	**K09338**	**Class III homeodomain-leucine zipper**
	Solyc10g006720.4.1	3.5	24.431	21	551	571	UCGGACCAGGCUUCAUUCCCC	CUGGAAUGAAGCUUGGGCGGA	Cleavage	K04733	G-type lectin S-receptor-like serine/threonine-protein kinase
	Solyc03g121640.3.1	3.5	13.824	21	795	814	UCGGACCAGGCUUCAUUCCCC	AAGGAAUGAAGCUUGG-CCGA	Cleavage	K04077	Chaperonin-60 kDa protein
** *miR6027-5p* **	Solyc07g047990.1.1	2	16.738	22	435	456	AUGGGUAGCACAAGGAUUAAUG	UCAUGAUCCUUGUGUUAUUCAU	Cleavage	K08867	MAP kinase kinase kinase 49
	Solyc09g064270.3.1	3	12.926	22	1783	1804	AUGGGUAGCACAAGGAUUAAUG	UUCUAAUCCUCGUGUUAUUCAU	Cleavage	K13430	Receptor-like serine/threonine-protein kinase ALE2
	**Solyc01g107670.2.1**	**3.5**	**14.541**	**22**	**213**	**234**	**AUGGGUAGCACAAGGAUUAAUG**	**CUCUGUUCCUCGUGUUACCCAU**	**Cleavage**	**NA**	**Leucine-rich repeat receptor-like protein kinase**
** *miR398-3p* **	Solyc02g078720.4.1	3.5	16.291	21	652	672	UGUGUUCUCAGGUUACCCCUG	AAAGGGUAACCUGAGCAUAUA	Cleavage	NA	Multidrug resistance protein
	**Solyc05g006630.4.1**	**3.5**	**22.484**	**21**	**558**	**578**	**UGUGUUCUCAGGUUACCCCUG**	**CUGGGGAAACUUGAUAAUACA**	**Cleavage**	**K19613**	**Disease-resistance-like protein (TIR-NBS-LRR class)**
** *novel miR1767* **	Solyc12g014490.3.1	2.5	15.25	24	1393	1416	CUUCAACUUUGGGUGUGCACAAGU	AGAGGUGCACACUUAAAUUUGAAG	Cleavage	K16732	Microtubule-associated protein MAP65-1c
	Solyc03g116760.3.1	3	19.108	24	1251	1274	CUUCAACUUUGGGUGUGCACAAGU	GAACUUGCAGACCCAAGGUUGAGU	Cleavage	K13416	LRR receptor-like serine/threonine-protein kinase FEI 1
	Solyc05g053260.3.1	3.5	18.48	24	199	222	CUUCAACUUUGGGUGUGCACAAGU	GUGGAAUCAUGCCUAAAGUUGAAG	Cleavage	NA	DNA (Cytosine-5)-methyltransferase DRM2
** *miR5300* **	**Solyc03g116550.4.1**	**2.5**	**17.866**	**22**	**936**	**957**	**UCCCCAGUCCAGGCAUUCCAAC**	**ACAGGAAACCUUGGACUGGGGA**	**Cleavage**	**NA**	**O-fucosyltransferase family protein (AT1G52630-like protein)**
	Solyc05g008650.1.1	3	20.456	22	1282	1303	UCCCCAGUCCAGGCAUUCCAAC	GUUGGAAUGCCUGGACUUGGCA	Cleavage	K13453	Late blight-resistance protein R1-A (NBS-coding resistance gene protein)
	Solyc06g064690.2.1	3	19.297	22	40	61	UCCCCAGUCCAGGCAUUCCAAC	UAUGGAAUGCCUGGACUUGGUA	Cleavage	K13453	NBS-coding resistance gene analog
** *miR6024* **	**Solyc10g051050.3.1**	**1**	**21.519**	**22**	**665**	**686**	**UUUUAGCAAGAGUUGUUUUACC**	**GGUAAGACAACUCUUGCUAGAA**	**Cleavage**	**K13453**	**Disease-resistance protein (AT4G27190-like protein)**
	Solyc11g065780.3.1	2.5	14.182	22	469	490	UUUUAGCAAGAGUUGUUUUACC	GGUAAGACAACACUUGCUAAAG	Translation	K15078	CC-NBS-LRR type resistance-like protein/Cc-nbs-resistance protein

## Data Availability

The sequence data were deposited in the NCBI Short Read Archive (SRA) with the Submission ID: SUB13960687, BioProject: PRJNA1037012, and Biosamples: SAMN38155784 to SAMN38155789.

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
