# Peer review of "Identification of Tomato microRNAs in Late Response to Trichoderma atroviride"

_ijms, 2024, doi:10.3390/ijms25031617_

Round 1

Reviewer 1 Report

Comments and Suggestions for Authors

In this manuscript Olmo and collaborators present an analysis aimed at identifying miRNA related in the priming of Trochoderma-mediated infection in tomato. The work is well described and well written and help shading some light in the phenomenon of Trichoderma protection.

There are some parts of the paper, however, that would deserve a better description and a more in-depth discussion, the most stricking being (as you would expect) the behavior of sample T11-R1 and, following, some results (validations indeed) regading the potential role of some of the miRNA analyzed in modulating the expression levels in the two conditions.

Getting into the text:

- you report that only a small amount of reads (uniquely) mapped onto tomato genome: is that usual? and what about the other reads? Do they map multiple times (that may have sense, if they target multiple genes) and/or which is the amount of unmapped reads? Could they refer to Trichoderma genome?

- T11-R1 shows a completely different behavior with respect to the other 2 treated samples. You suggest that the influence of Trichoderma affects certain specific functions, but if this is true, I guess that the profiles of the T11-treated samples would be similar: could it be that the treatment was somehow not as effective for sample 3?

When performing the analyses you considered two out of 3 samples to identify miRNAs: was it the case for the analyses performed in section 2.3 as well? you shoud comment on the reason

- you describe (line 125) miRNAs reads as being mainly 21 bp in length, but most of the miRNA of table S1 are 24bp long, and this difference may lead to misinterpretations, so that you should report that in the text. the fact tht a higher number of reads are 21bp long only suggests that such miRNAs are more expressed (other than perhaps false positives for 24bp-miRNAs). A low expression of some miRNA (as in the case of novel miR1767) may explain the differences you found in expression levels between sequencing and qPCR.

-line 189: please add (in brackets maybe) the novel miRNA identified only in control plants. This miRNA is interesting as potentially related to a resistance gene whose expression may be silenced in normal conditions.

- Figure 1A: y axis is not to scale according to what you report in the text; can you comment on this?

- Figure 1B: numbers do not match with those reported in the main text and in suppl tables (ex.: in S2 there are 87 putative miRNAs vs 88 and 72 instead of 73)

Author Response

Reviewer 1 report

In this manuscript Olmo and collaborators present an analysis aimed at identifying miRNA related in the priming of Trichoderma-mediated infection in tomato. The work is well described and well written and help shading some light in the phenomenon of Trichoderma protection.

There are some parts of the paper, however, that would deserve a better description and a more in-depth discussion, the most stricking being (as you would expect) the behavior of sample T11-R1 and, following, some results (validations indeed) regarding the potential role of some of the miRNA analyzed in modulating the expression levels in the two conditions.

We greatly appreciate Reviewer 1's comments. We have addressed all his/her concerns and comments, so we believe that the manuscript has substantially improved in quality.

- You report that only a small amount of reads (uniquely) mapped onto tomato genome: is that usual? and what about the other reads? Do they map multiple times (that may have sense, if they target multiple genes) and/or which is the amount of unmapped reads? Could they refer to Trichoderma genome?

We are very grateful for the reviewer's comment, since thanks to this input we have had the opportunity to present a more explanatory version of the table. The former field called “Total mapped reads” referred to the total number of reads that mapped against the S. lycopersicum genome and were considered for further downstream analysis. This included those reads that aligned only ONCE against the S. lycopersicum genome. The overall alignment rate was ca. 98%. However, about 80% of the reads aligned in more than one position in the genome (more likely due to the small length of the RNAseq library, which could lead to spurious alignments), so they were discarded for downstream analysis. The former field called “Unique mapped reads” referred to the number of reads resulting after one of the processing steps within the miR-PREFeR pipeline. We have decided to remove that field and show instead the number of reads that aligned with candidate miRNAs, as we believe the data would be more informative and clearer. We have reworded the text and legend to clarify the data and content of Table 1.

- T11-R1 shows a completely different behavior with respect to the other 2 treated samples. You suggest that the influence of Trichoderma affects certain specific functions, but if this is true, I guess that the profiles of the T11-treated samples would be similar: could it be that the treatment was somehow not as effective for sample 3?

We have removed the sentence in lines 132-134.

- When performing the analyses you considered two out of 3 samples to identify miRNAs: was it the case for the analyses performed in section 2.3 as well?

We thank the Reviewer for the comment, as this issue could be a matter of misinterpretation. The analyses were performed by considering all samples. To rule out possible false positives during miRNA prediction, these were considered only if they showed RNAseq reads aligned with them in two of the three samples per condition. Additionally, miRNA candidates that showed less than 10 reads aligning against their miRNA* sequence were also discarded. This situation only affects the prediction of miRNA candidates. All the other analyses, including differential expression, qPCR expression of the miRNA and validation of the targets were performed by using the three samples per condition. We have modified different sections of the main text (highlighted in yellow in section 2.1) to clarify this situation.

- you should comment on the reason you describe (line 125) miRNAs reads as being mainly 21 bp in length, but most of the miRNA of table S1 are 24bp long, and this difference may lead to misinterpretations, so that you should report that in the text. The fact that a higher number of reads are 21bp long only suggests that such miRNAs are more expressed (other than perhaps false positives for 24bp-miRNAs). A low expression of some miRNA (as in the case of novel miR1767) may explain the differences you found in expression levels between sequencing and qPCR.

Thanks again to the reviewer for this comment since the clarification of all these issues in the main text facilitates the readability of the manuscript. Table S1 contains the list of all potential miRNA identified (n=1889). From this list, only those miRNAs overcoming the threshold conditions (i.e. more than 10 reads aligning against the miRNA* sequence and transcription in at least two of the three samples per condition) were selected for further downstream analysis (highlighted in light gray in Table S1 and summarized in Table S2). The Reviewer is correct that most of the candidate miRNAs identified had a qualitative length of 24 nt. Quantitatively, however, when we considered the fraction of the RNAseq dataset that aligned with candidate miRNAs, we found that most reads aligned to 21- nt.-long miRNA, (as the Reviewer rightly says, 21- nt. miRNA were expressed more). This quantitative information is what we wanted to highlight in the main text and, therefore, we have modified the text accordingly in section 2.1 of the Results (“Reads of 20–24 nt long miRNAs accounted for over 99.99% of the total number of reads that aligned against miRNA candidates, among which 21 nt miRNA long reads were the ones that showed the greatest transcription overall, as they accounted for most of the sequencing reads (86.6% of all miRNA reads”) and the second paragraph of the Discussion to clarify this situation.

Regarding the 24-nt miRNAs, such as miR1767 and miR257 (both new), the reviewer is correct. The expression of these miRNAs was very low compared to the others listed in Table 2 (as can be seen in Table S2). However, we decided to include them as they were fulfilling the threshold applied to the others. In the case of miR1767, it was found in low expression in controls and absent in T11, thereby showing high DGE after DeSEQ2 statistical analysis. As the Reviewer points out, the differences between sequencing and qPCR might be due this very low expression found in the sequencing, and thus we have modified the section 2.2 (see the last paragraph) to highlight this fact.

- line 189: please add (in brackets maybe) the novel miRNA identified only in control plants. This miRNA is interesting as potentially related to a resistance gene whose expression may be silenced in normal conditions.

We agree with this suggestion. Therefore, we have included this information in the third paragraph of section 2.2.

- Figure 1A: y axis is not to scale according to what you report in the text; can you comment on this?

We strongly appreciate the reviewer's comment, as this figure could have been the cause of a misunderstanding. The numbers in the text refer to the percentage of 21nt (etc.) miRNA compared to the total number of reads that aligned against miRNA in S. lycopersicum. In Figure 1A, we wanted to highlight and display only those miRNAs that harbored at least 10 read counts against their miRNA* sequence. We have modified the legend and main text accordingly.

- Figure 1B: numbers do not match with those reported in the main text and in suppl tables (ex.: in S2 there are 87 putative miRNAs vs 88 and 72 instead of 73)

Thank you, we appreciate this notice. We have corrected the data in Figure 1B.

Reviewer 2 Report

Comments and Suggestions for Authors

There are some concerns about this paper:

1) It is not appropriate to talk about the data in supplementary materials in a whole section without data in the main manuscript (i.e., section 2.2). Please consider to move important information to the main manuscript.

2) We usually use p values < 0.05 as the threshold. Why authors use padj < 0.1? Please provide references if you think it is reasonable. line 172 and Table 2.

3) The information in lines 170-173 doesn't match with the information in the Abstract.

4) Are the values of log2FC  means of 3 replicates in Table 2? If they are, please provide standard errors.

5) Please provide standard errors in Fig. 3.

6) How do you define differentially expressed miRNAs? we usually consider the absolute values of log2FC at least >=1 as differentially expressed genes, i.e. two-fold change. Please provide references if you think your data is reasonable.

7) Lines 129-134, the description of PCA results is not understandable. If you have different results in replicates, suggesting it is not a universal result can be applied as plants' responses. Then, what is the significant value of the research? If you think it is reasonable, please provide references, and discuss it first in the Discussion section to convince readers.

8) The information of line 114-115 (one plant per replicate) is not consistant with the information provided in section 4.1 lines 135-136 (three plants per replicate).

9) Why did authors use 4-week-old or 30-day plants as your materials, please provide references?

10) The validation of miRNAs and their target genes using only qPCR is not enough to have your conclusions. 

Author Response

Reviewer 2 report

Dear Reviewer, we strongly appreciate your feedback. We have tried to address all your concerns.

There are some concerns about this paper:

1) It is not appropriate to talk about the data in supplementary materials in a whole section without data in the main manuscript (i.e., section 2.2). Please consider to move important information to the main manuscript.

The Reviewer is right. The results in section 2.2 are supported by data presented as supplementary material. Thus, to avoid giving relevance to a section only with supplemental material, we have combined sections 2.1 and 2.2, since the data shown in Tables S1 to S4 include a huge number of values and fields that hinders their inclusion within the main text. We thank the reviewer for this comment, as we consider that the changes performed in the main text according to this issue greatly increase the readability of the manuscript.

2) We usually use p values < 0.05 as the threshold. Why authors use padj < 0.1? Please provide references if you think it is reasonable. line 172 and Table 2.

All miRNA included in Table 2 showed p values (< 0.05) according to the Wand test, which were subsequently corrected following Benjamini and Hochberg’s method to assess False Discovery Rate (adjusted p value). Only one of the miRNAs in the study (miR398-3p) showed a p value <0.05 and padj>0.05 (padj=0.066). As the goal of miRNA selection in this part of the study was to choose potential candidates for further validation using other methods rather than sequencing and based on the recent results of López-Galiano et al. (2019, doi: 10.3390/genes10060475), where they also included candidates with padj < 0.1, we decided to keep miR398-3p for validation analyses. qPCR results agreed with the sequencing ones, so we kept miR398-3p in Table 2. We thank the Reviewer for the comment, and we have rephrased the corresponding M&M section 4.5 and the second paragraph of the Discussion (“Among the 72 preselected miRNAs, only 10 were DE (p adjusted < 0.1, according to [53] in T11-treated plants.”) to explain this issue.

3) The information in lines 170-173 doesn't match with the information in the Abstract.

We thank the reviewer for the notification. We have modified the main text (first paragraph of section 2.2) accordingly to match the information contained in the Abstract.

4) Are the values of log2FC means of 3 replicates in Table 2? If they are, please provide standard errors.

All samples from all conditions (three biological replicates per condition) were used for the calculation of DGE and Log2FC using DeSEQ2 and the unnormalized RNAseq counts aligned to the different miRNA candidates. Similarly, the log2FC standard error (lfcSE) was calculated and has been included in Table 2 according to the Reviewer's comment.

5) Please provide standard errors in Fig. 3.

As we have included the log2FC standard error (lfcSE) in Table 2, this may be sufficiently explanatory. In similar relevant articles, such as Lopez-Galiano et al., 2019; Cabrera et al., 2016; Luan et al., 2015, the authors included this type of graphics, without the standard errors when comparing trends between RNA-seq and qPCR miRNA data.

López-Galiano, M.J.; Sentandreu, V.; Martínez-Ramírez, A.C.; Rausell, C.; Real, M.D.; Camañes, G.; Ruiz-Rivero, O.; Crespo-Salvador, O.; García-Robles, I. Identification of stress associated microRNAs in Solanum lycopersicum by high-throughput sequencing. Genes 2019, 10, 475. [doi: 10.3390/genes10060475]

Cabrera, J., Barcala, M., García, A., Rio‐Machín, A., Medina, C., Jaubert‐Possamai, S., ... & Escobar, C. Differentially expressed small RNA s in Arabidopsis galls formed by Meloidogyne javanica: a functional role for miR390 and its TAS 3‐derived tasiRNAs. New Phytologist 2016, 209, 1625-1640. [doi:10.1111/nph.13249]

Luan, Y., Cui, J., Zhai, J., Li, J., Han, L., & Meng, J. High-throughput sequencing reveals differential expression of miRNAs in tomato inoculated with Phytophthora infestans. Planta 2015, 241, 1405-1416. [doi:10.1007/s00425-015-2267-7]

6) How do you define differentially expressed miRNAs? we usually consider the absolute values of log2FC at least >=1 as differentially expressed genes, i.e. two-fold change. Please provide references if you think your data is reasonable.

The Reviewer is correct and a minimum LFC of 1 is normally used to consider DE genes. However, the purpose of the DE analyses that resulted in the short list of 10 miRNA summarized in Table 2, was to highlight candidate miRNAs potentially DE in T11-treated and untreated plants, and to validate this potential DE by a molecular method other than RNAseq alone. Thus, we relied on the different information retrieved after sequencing, not only LFC, but also p-value, adjusted p-value, CPM values and transcription level (raw count number) of the different miRNA per sample and condition. After validation, we observed that qPCR values for highly transcribed miRNA (such as MIR166) were in line with the sequencing results and with a LFC > 1. Thus, we decided to keep all the miRNA that overcame the different thresholds (i.e. more than 10 reads aligning to miRNA* sequence, transcript values in at least 2 out of the three biological replicates per condition and DE between conditions with p value (wand) < 0.05 and p adj (BH) <0.1) in the main text and Table 2. To clarify our decision and in accordance with the Reviewer’s comment, we have made changes (highlighted in yellow in section 2.2) in this new version of the manuscript.

7) Lines 129-134, the description of PCA results is not understandable. If you have different results in replicates, suggesting it is not a universal result can be applied as plants' responses. Then, what is the significant value of the research? If you think it is reasonable, please provide references, and discuss it first in the Discussion section to convince readers.

We appreciate this comment as this section might have been a source of misinterpretation. We are very sorry because it looks clear that the message that we wanted to highlight was not well written. We have modified the section 2.1 of the Results. In addition, we have included in Figure S1 (in dark red) the restricted shortlist of 10 miRNAs that were selected for validation.

8) The information of line 114-115 (one plant per replicate) is not consistent with the information provided in section 4.1 lines 135-136 (three plants per replicate).

We have modified it accordingly in the first paragraph of Results, since we have used three biological replicates containing three plants each.

9) Why did authors use 4-week-old or 30-day plants as your materials, please provide references?

We have seen that the "Trichoderma effect" is transmitted to the offspring in terms of defense priming and growth promotion [Medeiros et al., 2017], so that in some way such effect must be kept by the plant throughout its life. Trichoderma-primed plants store the responses to multiple events in their ‘transcriptional memory’, modulating the transcription of response genes to future stress [Woo et al., 2023]. As indicated in the text (Lines 102-106), the priming effect is durable and effective in adapting to pathogen life cycles [Martínez-Medina et al., 2017; Medeiros et al., 2017], but the Trichoderma-triggered phytohormone signaling network alert disappears over time. We have used 30-day-old plant material because such alert becomes almost imperceptible at least four weeks after the plant has been in contact with the fungus, to save energy when not needed [Dominguez et al., 2016]. Previous works have analyzed the expression profiles of miRNAs in plants at very short times after Trichoderma application, while we have wanted to explore which miRNAs related to plant defense are differentially expressed when priming is dormant in plants, at a transcriptional memory state.

We have introduced this explanatory paragraph in the Discussion:

“Previous works have analyzed the expression profiles of miRNAs in plants at very short times after Trichoderma application [35, 37], while we have wanted to explore which miRNAs related to plant defense are differentially expressed when priming is dormant and only a few genes are differentially transcribed in Trichoderma-treated plants compared to the control [40], at a transcriptional memory state. Thus, we have performed RNA sequencing with three biological replicas of 30-day-old tomato leaf samples from untreated plants and plants from seeds coated with T. atroviride T11.”

10) The validation of miRNAs and their target genes using only qPCR is not enough to have your conclusions.

The reviewer is right. Indeed, it is risky to draw categorical conclusions without further functional analysis. Therefore, we have lightened the categorization of conclusions and constructed the statements with verbs in the subjunctive mood or in the continuous form.

We have modified the text as follows:

Tomato plants inoculated with T. atroviride T11 showed that the downregulation of miRNAs such as miR398, miR408, and miR6027, and the upregulation of miR6024, did not lead to modifications in the expression of their PRR or NLR targets, which would indicate that, pending more detailed functional studies on miRNA-target pairs, at a transcriptional memory state level the defences would be switched off. In addition, miRNAs showing canonical behavior with their respective targets: decrease of miRNA166 level and its upregulated target HD-ZIP class III and increase of miRNA5300 level and its downregulated target O-fucosyltransferase, could both be indicating that growth promotion in plants treated with T11 would be remaining active one month after inoculation of this beneficial fungus.”

Round 2

Reviewer 1 Report

Comments and Suggestions for Authors

I thank the authors for providing explanations for their interpretations. Some of them were indeed foreseeable, but I believe it is better to explain them to make the text understandable also for readers not familiar with certain arguments. Moreover, but this is my personal feeling, I feel that it is better to explain also unexpected or putatively contradictory results to be transparent and trustworthy.

I believe the authors have answered all my concerns so that the manuscript can be accepted as is; just add commas in Tables 1 and 3 to allow a better readability of numbers (ex.: 1,000,000 instead of 1000000).

Comments on the Quality of English Language

just some typos left (ex.: 30-days-old)

Author Response

Dear Editor,

Many thanks for your quick and positive response. We have made the small changes suggested by Reviewer 1, highlighted in blue shading.

With kind regards

Prof. Enrique Monte
